# ^18^FDG-PET/CT-Scans and Biomarker Levels Predicting Clinical Outcome in Patients with Alveolar Echinococcosis—A Single-Center Cohort Study with 179 Patients

**DOI:** 10.3390/pathogens12081041

**Published:** 2023-08-14

**Authors:** Lynn Peters, Wanjie Jiang, Nina Eberhardt, Jürgen Benjamin Hagemann, Beate Grüner, Dennis Tappe

**Affiliations:** 1Department of Internal Medicine III, Division of Infectious Diseases, Ulm University Hospital, 89081 Ulm, Germany; lynn.peters@uniklinik-ulm.de (L.P.); wanjie.jiang@uniklinik-ulm.de (W.J.); 2Department of Nuclear Medicine, Ulm University Hospital, 89081 Ulm, Germany; nina.eberhardt@uniklinik-ulm.de; 3Institute of Medical Microbiology and Hygiene, Ulm University Hospital, 89081 Ulm, Germany; juergen.hagemann@uniklinik-ulm.de; 4Bernhard Nocht Institute for Tropical Medicine, 20359 Hamburg, Germany

**Keywords:** alveolar echinococcosis, *Echinococcus multilocularis*, positron-emission tomography, ^18^FDG, soluble interleukin 2 receptor, cytokeratin fragments, Ck18F, IgE, Em2^+^, serology

## Abstract

**Background:** Alveolar echinococcosis (AE) is a severe larval tapeworm infection with a variable clinical course of the disease. Reliable imaging techniques and biomarkers are needed to predict the course of the disease. **Methods:** 179 AE patients that received PET/CT scans between 2008 and 2012 were retrospectively included. From stored blood samples taken on the day of the scan, levels of IgE, parasite-specific serology, amyloid A, C-reactive protein, soluble interleukin 2 receptor, cytokeratin fragments, eosinophilic cell count, and eosinophil cationic protein were measured. Additionally, the current clinical outcome (cured, stable, or progressive disease) after a median duration of 8 years after baseline examination was assessed. Ultimately, an ordinal logistic regression was conducted to evaluate which imaging parameters and biomarkers independently influence the clinical outcome. **Results:** In general, patients in need of medical treatment or with progressive disease, advanced PNM stages, and positive PET/CT scans exhibited higher levels of the respective biomarkers. However, only the parasite-specific serological markers and total IgE levels differed significantly between clinical groups, WHO PNM stages, and the results of the PET/CT scan. In the multivariate analysis, PET/CT results were a strong predictor of the clinical outcome (OR 8.908, 95%CI 3.019–26.285; *p* < 0.001), and age at baseline was a moderate predictor (OR 1.031, 95%CI 1.003–1.060; *p* = 0.029). **Conclusions:** The PET/CT scan is, preferably in combination with parasite-specific serology and IgE levels, a valuable tool in the clinical management of AE and is able to predict the course of the disease.

## 1. Introduction

Alveolar echinococcosis (AE), an infection with the larval stage (metacestode) of the fox-tapeworm *Echinococcus multilocularis*, is an emerging parasitic zoonosis of the northern hemisphere [1,2]. AE mainly affects the liver, and the metacestode tissue grows infiltratively and can metastasize [3,4]. The disease is staged according to the World Health Organization (WHO)-PNM system (P = location of the parasitic mass in the liver, N = involvement of neighboring organs, M = metastases), which translates to the WHO stages I, II, IIIa, IIIb, and IV, based on the extent of the lesion(s) and the structures or organs involved [5,6]. In the early stages, curative surgery is the treatment of choice. However, the majority of patients present in advanced stages with involvement of the hepatic hilus and/or infiltration of neighboring organs and/or distant metastasis, hence complete resection of all parasitic lesions cannot be achieved [7,8]. In these cases, the only therapeutic option is long-term anthelminthic treatment with benzimidazoles (BMZ).

Especially in inoperable patients, tools and markers are needed to reliably predict and differentiate a stable course from a progressive course of disease to guide individual treatment. For patients with inactive disease, a structured treatment interruption (STI) of the BMZ therapy can be a goal, not only to save costs but also to increase the quality of life. For those with progressive disease, i.e., growing lesions or the occurrence of AE-related complications, treatment should be intensified; in certain cases, rescue surgery needs to be discussed [7].

Hitherto, imaging techniques such as ultrasound (US), computed tomography (CT), and magnetic resonance imaging (MRI) are applied to assess the dynamic of the lesion’s extent. However, metacestode growth is slow, and proof of progressive disease requires several months [9]. Additionally, biomarkers including serology and systemic cytokine and chemokine levels are used in clinical practice as surrogate parameters to monitor the treatment response, since they correlate not only with the PNM stage but also with the extent of the lesion and the disease’s dynamic over time [9,10,11,12,13,14]. However, the number of prognostic biomarkers is limited, and reliable parameters are needed for the clinical management of AE.

In recent years, [18F]-fluorodeoxyglucose positron-emission tomography with computed tomography (FDG-PET/CT) has increasingly been used to evaluate disease activity. PET-positivity presumably reflects immune cell activity at the parasite–host interface, and thus disease activity [15]. It is assumed that increasing activity precedes disease progression and, vice versa, decreasing activity reflects treatment response. The lack of PET-positivity might indicate an inactive disease. However, the result of a PET/CT should always be viewed in context with other biomarkers and clinical findings. Data from Hotz et al. (2022) revealed strong correlations for both *Echinococcus* IgG and recEm18 serology with FDG tracer uptake in PET/CT scans [16]. Recent studies combining FDG-uptake by PET with specific serology (mainly Em18-ELISA) showed promising results regarding the safety of an STI [17,18].

Our study aimed to identify biomarkers corresponding to the extent and activity of AE as well as the clinical outcome. More specifically, we assessed the concentration of inflammatory biomarkers (serum amyloid A (SAA), C-reactive protein (CRP), and soluble interleukin 2 receptor (sIL-2R)), fibrosis/apoptosis biomarkers (cytokeratin fragments Ck18F (M30) and Ck18F (M65)), the eosinophil response (total eosinophilic cell count (EOS) and eosinophil cationic protein (ECP)), and levels of serological biomarkers (total immunoglobulin E (IgE total), parasite-specific IgE (specific IgE), indirect hemagglutination (IHA) IgG titers against a crude parasite extract, as well as antibody levels against Em2^+^, a specific parasite antigen combination). The levels of the respective biomarkers were related to the WHO PNM stage of AE, the PET/CT result, and the clinical outcome, i.e., if during follow-up a patient was defined as cured, stable with or without treatment, or had progressive disease.

## 2. Materials and Methods

### 2.1. Study Design

This study was designed as a retrospective cohort study. Every patient that received a routine PET/CT scan at our specialized outpatient department from 03/2008–12/2012 was included (*n* = 179). On the same day, blood samples were taken and stored for further analysis. Subsequently, the biomarkers described in the following sections were measured from the stored samples. The clinical outcome was assessed based on the most recent patient visit (07/2010–06/2022), which is documented in the electronic patient management system used in our hospital. Patients without a follow-up visit or with a follow-up period of <6 months (*n* = 20) were excluded from the analysis regarding clinical outcomes. Patients that had been operated on and no longer had a detectable lesion in the PET/CT scan were excluded from the analysis regarding PET/CT results (*n* = 39).

### 2.2. Patient Cohort

For this study, all AE patients who received a routine FDG-PET/CT scan during the study period (*n* = 179) were retrospectively included. At our center, FDG-PET/CT scans are performed as staging examinations and subsequently every two to three years as regular follow-up examinations, irrespective of the clinical status and the type of treatment. Scans with detectable metabolic activity were labeled ‘positive’ while those without were labeled ‘negative’.

Blood samples from all patients were routinely collected on the day of the PET/CT scan and analyzed for cell count and serology. The remaining serum samples were stored for further analysis at −20 °C and tested subsequently for the biomarkers described in the following sections.

Upon every follow-up visit thereafter, patients underwent clinical examination, routine laboratory testing, and imaging such as US, CT, or MRI scans. Based on the physician’s findings, a patient was classified as cured (no residual or reoccurring disease after surgical resection), stable disease (no curative surgery and no significant dynamic of the disease) without BMZ treatment, stable disease with BMZ treatment, or progressive disease (no curative surgery and growing lesions or occurrence of AE-related complications, e.g., cholangitis, portal vein thrombosis). The clinical status was documented after every visit.

At our center, there are a number of patients with stable diseases not requiring mediation who, strictly speaking, do not fulfill the STI criteria proposed previously [7], namely a negative PET/CT scan, a negative or at least decreased specific serology, the absence of symptoms, and continuous treatment for at least two years.

The reasons for treatment discontinuation in these non-operable patients were manifold. The lesion was considered inactive in some cases; however, the majority of patients within this group either did not wish to continue treatment (patient-initiated treatment interruption) or had shown toxicity or side effects to earlier BMZ treatment, which led to treatment discontinuation. In these cases, the necessity for anthelminthic treatment must be weighed with the patient’s wishes or side effects. If the disease has shown treatment response earlier and there is no imminent risk for complications in the case of a progression, treatment discontinuation can be evaluated even though the STI criteria are not met. However, this decision can only be taken following an interdisciplinary board discussion, and the patient as well as the relatives need to be part of the decision-making process. Subsequently, close monitoring is needed in these patients.

This study was approved by the ethical committee of the University of Ulm (No. 372/15).

### 2.3. Serological Detection of Anti-Echinococcus Antibodies, Total and Parasite-Specific IgE, and Eosinophilic Cationic Protein

At the time of the study, serological testing for anti-echinococcal antibodies was performed at the Institute of Medical Microbiology and Hygiene at Ulm University Hospital. *Echinococcus* IgG was measured using the Cellognost^®^ *Echinococcus* IHA screening test (Siemens Healthcare Diagnostics GmbH, Marburg, Germany). Results were interpreted as positive if titers were >1:32. For this study, *Echinococcus* IHA results were recorded as semiquantitative values. Additionally, the Em2^+^ ELISA (Bordier Affinity Products SA, Crissier, Switzerland) was performed and interpreted qualitatively as positive according to the manufacturer’s instructions if the index $\frac{absorption\;of\;patient\;serum\;sample}{absorption\;of\;cut-off\;control}$ was ≥1.0. All ELISA washing steps were performed with an automated microplate washer (Biochrom Ltd., Cambridge, UK), and absorbance measurements were carried out on an Infinite F50 microplate reader (Tecan Group Ltd., Männedorf, Switzerland). All patient sera were routinely further analyzed for their total IgE levels, using an electrochemiluminescence immunoassay on a Cobas e801 platform (Roche Diagnostics Deutschland GmbH, Mannheim, Germany), their *Echinococcus*-specific IgE fraction, and the eosinophilic cationic protein (ECP) using the ImmunoCAP^250^ (Phadia AB, Freiburg, Germany).

### 2.4. Measurement of sIL-2R, Ck18F, SAA, CRP and EOS

Frozen blood samples from the baseline visit were used for the detection of sIL-2R (Immulite 1000, Siemens, Nuremberg, Germany; upper reference limit of 710 U/mL), cytokeratin fragments (Ck18F-M30 and Ck18F-M65; PEVIVA M30 and M65 ELISAs, VLVbio, Nacka, Sweden; reference values of <260 U/L and <266 U/L, respectively), and SAA and CRP levels (clinical routine; upper reference limits of <10 mg/L and <5 mg/L respectively).

Eosinophilic cells were counted in an automated differential blood count (Sysmex Deutschland GmbH, Norderstedt, Germany).

### 2.5. PET/CT Study

For PET/CT examinations, patients were scanned with either the PET/CT scanner Discovery LS from General Electrics (GE) or with the Biograph mCT-S (40) from Siemens. The acquisition of the data with the Discovery LS scanner was in 2 D PET mode and for the Biograph mCT-S in full 3 D mode with time-of-flight (TOF) and point spread function corrections. All scan protocols were carried out from the skull base to the groin for PET and CT data with a measuring time per PET bed position of 2.5 min. For PET, 18F-FDG was administered intravenously as a tracer to visualize the glucose uptake approximately one hour before the scan. Patients had to fast at least for four to six hours prior to injection, and blood glucose levels measured before injection were allowed up to 180 mg/dL or 10.0 mmol/L. The injected doses of FDG were adapted by body weight (up to a total of 310 to 340 MBq for the Biograph mCT-S and up to 350 to 550 MBq for the Discovery LS according to former diagnostic reference values). In total, 10 mg of furosemide was injected for faster renal clearing of FDG. CT scans were performed in a supine position with arms overhead whenever possible right before PET either without or, where possible, with an intravenous contrast agent (Ultravist-300, Bayer AG, Leverkusen, Germany), weight-adapted 1.5 mL/kg body weight (min. 60 mL, max. 120 mL). The CT scans were acquired in shallow expiration after careful instruction of the patients. PET and CT images were viewed separately and as PET/CT fusion images on a Siemens Syngo^®^ Via Workstation and the current Picture Archiving and Communication System (PACS).

Image analysis was performed by at least one nuclear medicine specialist. For this study, we only categorized scans visually having FDG uptake around the AE lesions above the background liver uptake/non-affected liver tissue (PET positive) or not/AE lesion with no detectable FDG uptake (PET negative).

### 2.6. Statistical Analysis

Statistical analysis was performed using IBM SPSS version 28. Demographic data and significant results were described with the mean (M), median (MD), standard deviation (SD), interquartile range (IQR) for continuous and percentiles (P_25_, P_75_) for ordinal data, and minimum (Min) and maximum (Max) values. The Shapiro–Wilk test showed a non-normal distribution of the data of interest. Therefore, continuous and ordinal data were analyzed using the Mann–Whitney-U-test (*U*); categorical variables were analyzed using the Chi-square test (*Χ*^2^). Differences across groups were assessed using the Kruskal–Wallis-test (*Χ*^2^). Correlations were performed according to Spearman (*r_s_*). Missing data were handled by listwise exclusion for the univariate analysis and by multiple imputation for the multivariate analysis [19]. To evaluate the influence of different variables on the clinical outcome while adjusting for confounders and interactions between variables, an ordinal logistic regression analysis was conducted. Multicollinearity between variables was excluded. Results were considered significant with a *p*-value smaller than 5%.

## 3. Results

To keep this section concise, only the most important and significant findings are presented. For all detailed results, we refer to Appendix A.

### 3.1. Demographic and Descriptive Results

Of the 179 patients included, 76 (54%) were female. The mean age was 57 years (SD = 17, MD = 60, Min = 17, Max = 94).

Only 3 patients presented in WHO stage I (1.7%), while 34 (19.0%) presented in stage II, 48 (26.8%) in stage IIIa, 43 (24.0%) in stage IIIb, and 50 (27.9%) in stage IV.

Of the 179 patients, 28 presented at our center for the first time. A total of 17 patients lacked a follow-up visit and 3 patients received the follow-up visit within 6 months after the initial visit. The remaining 159 patients were followed up for a mean time of 7.2 years (SD = 3.2, MD = 8.0, IQR = 5.0).

Of these patients, 37 (23.3%) were considered cured. Most patients were defined as ‘stable’ at the end of the follow-up period, of which 76 (47.8%) were on BMZ therapy and 39 (24.5%) did not need medical treatment. In total, 7 (4.4%) patients had a progressive disease.

### 3.2. Interrelations between Biomarkers

There was a significant correlation between the levels of SAA and the inflammation markers CRP (*r*_s_ = 0.318; *p* < 0.001) and sIL-2R (*r*_s_ = 0.321; *p* < 0.001). The CRP levels correlated with the ECP levels (*r*_s_ = 0.257, *p* = 0.002). Levels of IgE and parasite serology correlated closely (specific IgE and IHA: *r*_s_ = 0.467; *p* < 0.001; specific IgE and Em2^+^: *r*_s_ = 0.523; *p* < 0.001; total IgE and IHA: *r*_s_ = 0.445; *p* < 0.001; total IgE and Em2^+^: *r*_s_ = 0.542; *p* < 0.001). Total and specific IgE levels correlated with EOS counts (*r*_s_ = 0.238; *p* = 0.005; *r*_s_ = 0.200; *p* = 0.019), while total IgE levels correlated with sIL-2R levels (*r*_s_ = 0.184; *p* = 0.035).

### 3.3. Biomarkers in Relation to PET/CT Results

There was no significant difference between patients with positive and patients with negative PET scans regarding SAA levels (U = 1357.50; *p* = 0.234), Ck18F(M30) concentrations (U = 1400.00; *p* = 0.636), Ck18F(M65) levels (U = 1255.50; *p* = 0.201), the M30:M65 ratio (U = 1270.00; *p* = 0.231), ECP levels (U = 1426.00; *p* = 0.741) and EOS count (U = 1469.00; *p* = 0.200). Although patients with positive PET scans showed slightly higher levels of CRP (MD = 2.50, IQR = 4.90 vs. MD = 1.85, IQR = 5.18; U = 1361.00; *p* = 0.489) and sIL-2R (MD = 264.00, IQR = 177.50 vs. MD = 228.00, IQR = 178.00; U = 1415.00, *p* = 0.489), these results were not statistically significant.

Regarding serological markers, patients with positive PET scans showed significantly higher levels of total IgE (U = 785.50; *p* < 0.001), specific IgE (U = 995.00; *p* < 0.001), IHA IgG (U = 1058.50; *p* = 0.001), and Em2^+^ IgG (*X*^2^(1) = 18.511; *p* < 0.001). The detailed results are summarized in Table 1 and depicted in Figure 1 and Figure 2. We provide some examples of positive and negative PET/CT scans in Figure 3.

### 3.4. Biomarkers and PET/CT Results in Relation to WHO Stage of Disease 

Patients with stage IV disease showed lower levels of SAA than patients in other WHO stages (MD = 3.20 mg/L vs. MD = 2.50–3.10 mg/L), but this result was not significant (*Χ*^2^(3) = 1.148; *p* = 0.765). Neither CRP nor sIL-2R concentrations differed significantly between WHO stages (CRP: *Χ*^2^(3) = 0.952; *p* = 0.813; sIL-2R: *Χ*^2^(3) = 0.952; *p* = 0.813). However, levels of Ck18F(M30) increased with advanced stages: While patients in stages I and II showed the lowest levels (M = 133.28 U/L, SD = 63.73, MD = 111.65, IQR = 63.20), patients in stage III showed slightly higher levels (IIIa: M = 138.17 U/L, SD = 61.80, MD = 128.69, IQR = 49.28; IIIb: M = 152.45 U/L, SD = 80.97, MD = 132.86, IQR = 64.94) and those in stage IV showed the highest concentrations (M = 197.20 U/L, SD = 235.00, MD = 132.90, IQR = 97.90). The result almost reached the significance level (*Χ*^2^(3) = 7.320; *p* = 0.062). Ck18F(M65) levels did not differ significantly between the clinical stages (*Χ*^2^(3) = 4.496; *p* = 0. 0.213), nor did the ratio of M30:M65 (*Χ*^2^(3) = 2.262; *p* = 0.520). EOS counts did not differ significantly between stages of disease (*Χ*^2^(3) = 2.630; *p* = 0.452), nor did ECP levels (*Χ*^2^(3) = 2.630; *p* = 0.452), in spite of a subtle tendency of the latter to increase with the stage of disease (MD_StageI_ = 2.85 µg/L to MD_StageIV_ = 3.60 µg/L).

The median level of both total IgE (*Χ*^2^(3) = 14.137; *p* = 0.003) and specific IgE (*Χ*^2^(3) = 10.442; *p* = 0.015) increased significantly with the WHO stage of the disease. The significance across groups is mainly attributable to the high median levels of IgE in patients with stage IV disease, which becomes apparent in Figure 4. Similarly, the proportion of positive Em2^+^ IgG increased significantly with higher stages (*Χ*^2^(3) = 16.472; *p* < 0.001) (Figure 5B). The level of IHA titers differed significantly across groups (*Χ*^2^(3) = 13.705; *p* = 0.003), which is mainly due to high levels in patients with stage IV disease (Figure 5A).

Regarding PET/CT results, there was a significant difference between the stages of disease regarding PET positivity (*X*^2^(3) = 10.346; *p* < 0.016). In all stages, the majority of patients showed positive PET/CT scans (Figure 6). However, the more advanced the stage of disease, the higher the proportion of patients with positive PET/CT scans. The details are displayed in Table 2.

### 3.5. Biomarkers and PET/CT Results in Relation to Clinical Outcome

SAA levels did not differ significantly between cured patients, those with stable disease with and without BMZ treatment, and those with progressive disease (*X*^2^(3) = 0.796; *p* = 0.850). Neither CRP nor sIL-2R concentrations differed significantly between outcome groups (CRP: *X*^2^(3) = 1.637; *p* = 0.651; sIL-2R: *X*^2^(3) = 0.621; *p* = 0.892). Ck18F(M30) levels did not differ significantly between different outcomes (*X*^2^(3) = 1.588; *p* = 0.662), and neither did Ck18F(M65) concentrations (*X*^2^(3) = 2.960; *p* = 0.398) nor the ratio of M30:M65 (*X*^2^(3) = 1.512; *p* = 0.680). EOS counts did not differ significantly between clinical outcomes (*X*^2^(3) = 4.596; *p* = 0.204), nor did ECP levels (*X*^2^(3) = 0.281; *p* = 0.964).

In contrast, the total and specific IgE levels differed significantly between outcome groups, with higher levels in patients requiring medication and with progressive disease compared to cured patients and those without the need for treatment (*X*^2^(3) = 40.856; *p* < 0.001; *X*^2^(3) = 38.564; *p* < 0.001; c.f. Table 3 and Figure 7A,B). The number of positive Em2^+^ IgG and the level of IHA IgG increased similarly with the outcome group (Table 3 and Figure 8A,B). These results were significant (*X*^2^(3) = 44.805; *p* < 0.001; *X*^2^(3) = 27.233; *p* < 0.001).

Furthermore, all patients with progressive disease were PET-positive, as well as the vast majority of patients that were stable with BMZ treatment. Of all patients with negative PET/CT scans, 70.1% were stable without BMZ treatment. Focusing on this group, approximately half of the patients showed a positive and the other half a negative PET/CT scan (Table 3 and Figure 9). Cured patients, e.g., after surgical resection, were excluded from this analysis.

A multivariate model was established to assess the influence of the different biomarkers and the PET/CT results on the clinical outcome while controlling for effect modifiers such as gender, age, and stage of disease (c.f. Table 4). The outcome categories were 1 = stable disease without medication, 2 = stable disease with medication, and 3 = progressive disease. Operated patients were excluded from the analysis. The overall model was significant compared to the null model (*X*^2^(12) = 34.105, *p* < 0.001) with appropriate goodness of fit (*X*^2^(190) = 198.630, *p* = 0.319) that explained 35.1% of the outcome’s variance (*R*^2^), which is a moderate proportion. The regression coefficients (*B*) that predict the change in log odds were transformed to odds ratios (*ExpB*) to facilitate the interpretation of the results.

The two variables significantly predicting the clinical outcome several years later were the age of the patient at baseline and a positive PET/CT result. Showing a positive PET/CT scan compared to a negative PET/CT scan increased the odds 8.908-fold of a patient being classified in a higher outcome category as opposed to a lower outcome category. In other words, this indicates that patients with positive PET/CT scans were more likely to require medication to remain stable or to develop a progressive disease over time. On the other hand, patients with negative PET/CT scans were more likely to remain stable without the need for anthelminthic treatment. Regarding age, the odds of a patient being classified in a higher outcome category compared to a lower outcome category increased 1.031-fold with every year of age. Hence, the probability of requiring medication to remain stable or developing progressive disease increases slightly with the age of a patient. Conversely, younger patients were more likely to achieve stable disease without the need for medical treatment.

## 4. Discussion

In this study, we investigated different biomarkers and the PET status of AE patients with regard to the clinical outcome and the WHO PNM stage to evaluate their benefit in clinical practice and to optimize the medical care of AE patients. Generally, in higher AE stages and in advanced clinical status, most biomarkers exhibited higher levels with varying significance.

Close correlations between different biomarkers were found in our study. As expected, correlations among general inflammation makers tested here (SAA, CRP, and sIL-2R), and correlations among serological tests (total IgE, the parasite-specific IgE fraction, anti-*Echinococcus* IHA IgG titers, and Em2^+^ status) were detected. Interestingly, correlations between CRP and ECP, between total and specific IgE and EOS counts, between total IgE and sIL-2R concentrations, and between CRP with ECP levels were also found. These results illustrate a fraction of the complex host response to AE involving interconnections between general inflammation, serological response, and eosinophil activation, which require further investigation.

Regarding the WHO PNM stages, no significant differences were seen for concentrations of the biomarkers SAA, CRP, sIL-2R, Ck18F(M65), the M30:M65 ratio, EOS counts, and ECP. However, almost significant increases were found for Ck18F(M30) with increasing PNM stage. As advanced stages are often caused by large hepatic lesions, this result might reflect the level of apoptosis by quantifying the concentration of caspase-cleaved keratin 18 (ccK18) released from epithelial cells. Similar to other hepatic disorders [20,21], this biomarker may be helpful in assessing the fibrosis grade of an AE-affected liver.

Total IgE, parasite-specific IgE, IHA IgG titers, and Em2^+^ status changed significantly with increasing PNM stage. A respective correlation of serology (crude antigen preparations or the recombinant/purified antigens EM10, Em18, and Em2^+^ antigens) and the PNM stage has been shown previously [14,16,17,18] and was confirmed in our investigation.

Between the clinical outcome groups (cured, stable with or without BMZ treatment, and progressive disease), no significant differences were found for levels of SAA, CRP, sIL-2R, Ck18F(M30), Ck18F(M65), the M30:M65 ratio, EOS counts, and ECP. However, significant increases were seen for total IgE, parasite-specific IgE, IHA IgG titers, and Em2^+^ status with advancing clinical status. For these biomarkers, significantly higher levels were also demonstrated in patients with positive versus negative PET results, similar to earlier studies [16].

The major finding of this study is that the PET/CT result at baseline was an independent and significant predictor of the clinical outcome assessed after a median follow-up period of 8 years. PET positivity is regarded as increasing disease activity, which is supposed to precede the growth of a lesion. A PET/CT scan at baseline should hence be implemented in clinical practice for the early identification of patients at risk. Conversely, a negative PET/CT result might indicate an inactive lesion that no longer requires anthelminthic treatment. Recently, Husmann et al. demonstrated that the quantitative activity assessed by PET/CT scans correlated with the duration of treatment and the time to serological negativity [22]. Earlier, the combination of negative PET/CT scans and anti-EmII/3–10 antibody levels were shown to be reliable parameters for assessing in vivo AE-larval inactivity after long-term BMZ therapy [17]. Furthermore, interrupting BMZ treatment in inoperable but PET-negative and serologically inactive AE was considered feasible with a favorable outcome [23]. However, prior research indicated that an STI based on a negative PET/CT only without taking serological parameters into account might result in relapsing disease [24]. Hence, the combination of PET/CT and biomarkers is required for the follow-up of patients with unresectable AE lesions. Nevertheless, the results of the study emphasize the importance of the PET/CT scan in the clinical management of AE patients, allowing a stratification with regard to the future outcome. Patients with positive PET/CT scans, especially with high perilesional activity, should be monitored closely. Hence, a PET/CT scan should be routinely offered to all AE patients as a staging examination and preferably as a follow-up imaging technique.

The strengths of this study were its considerable sample size in comparison with previous research regarding AE and the long-term observation of patient outcomes. However, our findings result from a retrospective study, and prospective research is needed to further investigate the relationship between PET/CT scans—preferably using quantitative measurements, biomarkers, and clinical characteristics.

## Figures and Tables

**Figure 1 pathogens-12-01041-f001:**
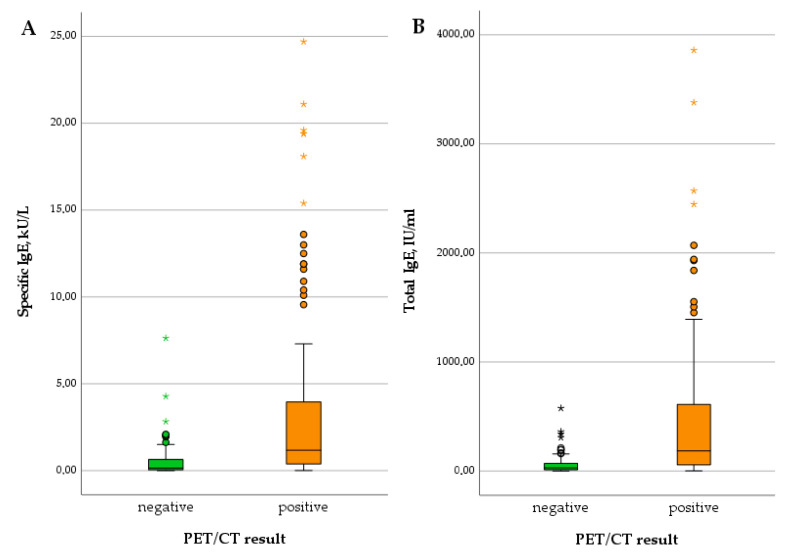
Levels of specific IgE (**A**) and total IgE (**B**) in patients with negative versus positive PET scan.

**Figure 2 pathogens-12-01041-f002:**
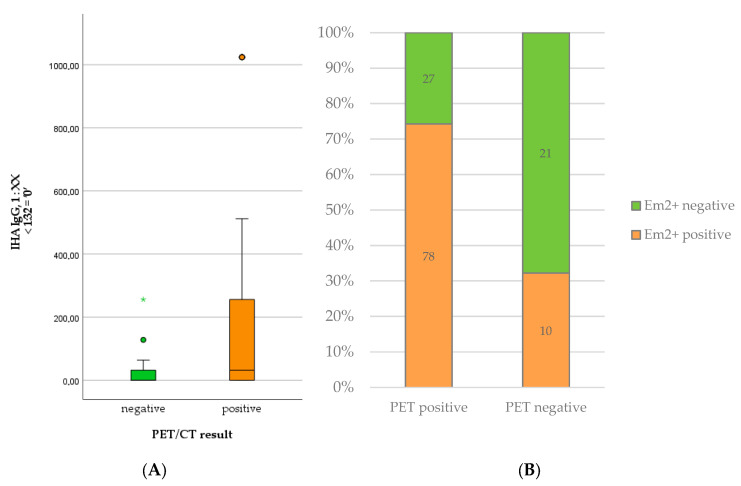
Levels of IHA IgG (**A**) and number of positive Em2^+^ assay results (**B**) in patients with positive versus negative PET scan.

**Figure 3 pathogens-12-01041-f003:**
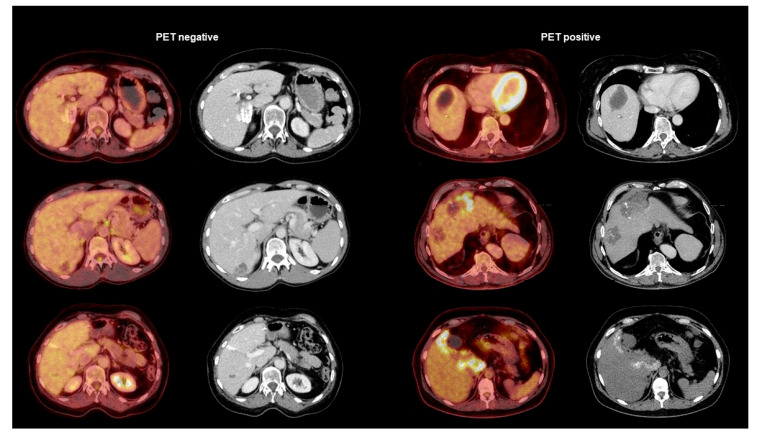
Examples of PET-negative AE lesions (**left side**) and PET-positive AE lesions (**right side**). The pictures show in each category the PET/CT fusion image on the left and the contrast-enhanced CT scan on the right.

**Figure 4 pathogens-12-01041-f004:**
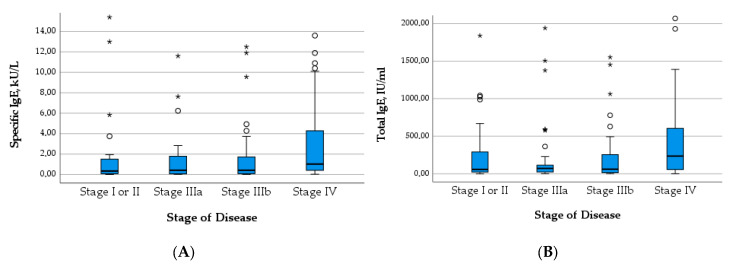
Levels of specific IgE (**A**) and total IgE (**B**) in relation to stage of disease.

**Figure 5 pathogens-12-01041-f005:**
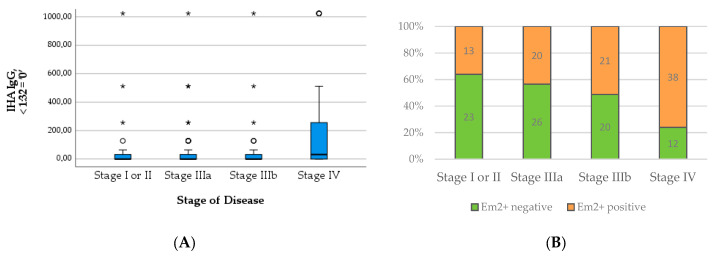
Levels of IHA IgG (**A**) and numbers of positive Em2^+^ assay result (**B**) in relation to stage of disease.

**Figure 6 pathogens-12-01041-f006:**
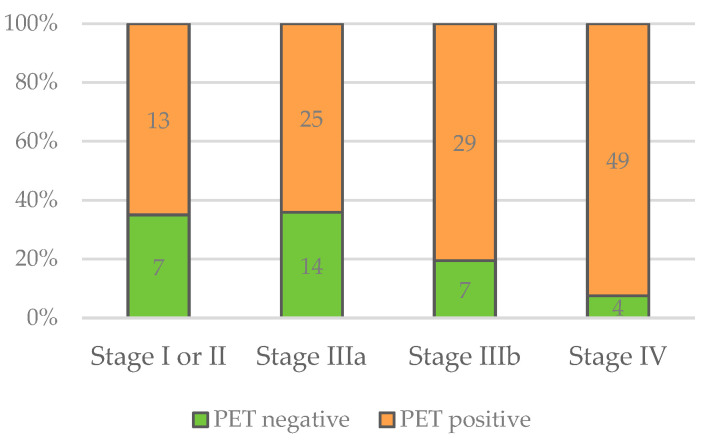
Numbers of negative and positive PET/CT scans in relation to stage of disease.

**Figure 7 pathogens-12-01041-f007:**
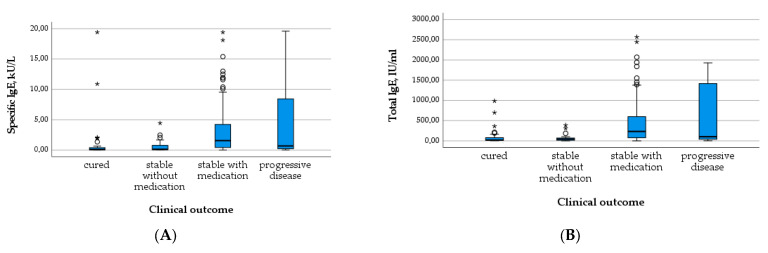
Levels of specific IgE (**A**) and total IgE (**B**) in relation to clinical outcome.

**Figure 8 pathogens-12-01041-f008:**
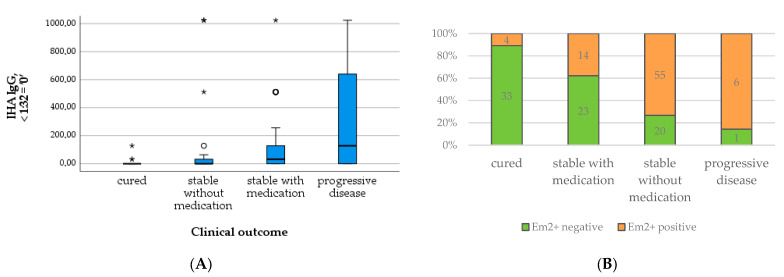
Levels of IHA IgG (**A**) and number of positive Em2^+^ assay results (**B**) in relation to clinical outcome.

**Figure 9 pathogens-12-01041-f009:**
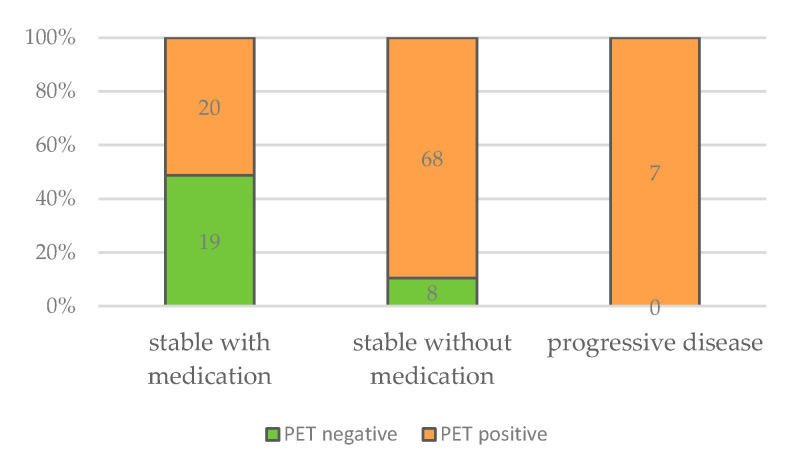
Numbers of negative and positive PET/CT scans in relation to clinical outcome.

**Table 1 pathogens-12-01041-t001:** Levels of specific and total IgE, IHA IgG, and N patients with/without Em2^+^ in relation to PET scan results.

	PET Scan		
Biomarker	Positive	Negative	Statistical Analysis
Specific IgE, kU/L	*n* = 107	*n* = 33	Mann-Whitney-U-test: *U* = 995.00; *p* < 0.001
	M = 4.11 SD = 7.44 MD = 1.18 Min = 0.01 Max = 49.10	M = 0.85 SD = 1.53 MD = 0.17 Min = 0.02 Max = 7.63	
Total IgE, IU/mL	*n* = 107	*n* = 33	Mann-Whitney-U-test: *U* = 785.50; *p* < 0.001
	M = 706.05 SD = 1637.33 MD = 185.30 Min = 1.14 Max = 14,089.00	M = 79.24 SD = 119.85 MD = 46.30 Min = 1.10 Max = 576.30	
IHA IgG, <1:32 = ‘0’	*n* = 105	*n* = 31	Mann-Whitney-U-test: *U* = 1058.50; *p* = 0.001
	MD = 32 P_25_ = 0 P_75_ = 256 Min = 0 Max = 8292	MD = 0 P_25_ = 0 P_75_ = 32 Min = 0 Max = 256	
Em2^+^ positive	*n* = 78 (72.9%)	*n* = 10 (30.3%)	Chi-Square-Test *Χ*^2^(1) = 18.511; *p* < 0.001
Em2^+^ negative	*n* = 27 (25.2%)	*n* = 21 (63.6%)	

**Table 2 pathogens-12-01041-t002:** Levels of specific and total IgE, level of IHA IgG, N patients with/without Em2^+^, and N patients with positive/negative PET/CT scans with regard to stage of disease.

	WHO Stage of Disease				
	Stage I and II	Stage IIIa	Stage IIIb	Stage IV	Statistical Analysis
Specific IgE, kU/L	*n* = 37	*n* = 48	*n* = 42	*n* = 50	Kruskal-Wallis-test: *Χ*^2^(3) = 10.442; *p* = 0.015
	M = 3.52 SD = 7.77 MD = 0.35 Min = 0.00 Max = 34.40	M = 2.01 SD = 4.29 MD = 0.43 Min = 0.02 Max = 21.10	M = 1.68 SD = 3.02 MD = 0.43 Min = 0.01 Max = 12.50	M = 4.24 SD = 8.07 MD = 1.03 Min = 0.03 Max = 49.10	
Total IgE, IU/mL	*n* = 37	*n* = 48	*n* = 42	*n* = 50	Kruskal-Wallis-test: *Χ*^2^(3) = 14.137; *p* = 0.003
	M = 608.03 SD = 2311.86 MD = 57.0 Min = 1.10 Max = 14,089.00	M = 199.75 SD = 398.43 MD = 71.75 Min = 2.20 Max = 1941.00	M = 272.00 SD = 518.11 MD = 61.25 Min = 1.14 Max = 2570.00	M = 755.64 SD = 1289.44 MD = 235.20 Min = 2.20 Max = 5750.00	
IHA IgG, <1:32 = ‘0’	*n* = 35	*n* = 46	*n* = 41	*n* = 50	Kruskal-Wallis-test: *X*^2^(3) = 13.705; *p* = 0.003
	MD = 0 P25 = 0 P75 = 32 Min = 0 Max = 1024	MD = 0 P25 = 0 P75 = 40 Min = 0 Max = 2048	MD = 0 P25 = 0 P75 = 32 Min = 0 Max = 8192	MD = 32 P25 = 0 P75 = 256 Min = 0 Max = 4096	
Em2^+^ negative	*n* = 23 (62.2%)	*n* = 26 (54.2%)	*n* = 20 (47.6%)	*n* = 12 (24.0%)	Chi-Square-test: *Χ*^2^(3) = 16.472; *p* < 0.001
Em2^+^ positive	*n* = 13 (35.1%)	*n* = 20 (41.7%)	*n* = 21 (50.0%)	*n* = 38 (76.0%)	
PET negative	*n* = 7 (35.0%)	*n* = 14 (35.9%)	*n* = 7 (19.4%)	*n* = 4 (9.1%)	Chi-Square-test: *X*^2^(3) = 10.346; *p* < 0.016
PET positive	*n* = 13 (65.0%)	*n* = 25 (64.1%)	*n* = 29 (80.6%)	*n* = 40 (90.9%)	

**Table 3 pathogens-12-01041-t003:** Levels of specific and total IgE and N patients with/without Em2^+^ with regard to clinical outcome.

	Clinical Outcome				
	Cured	Stable without BMZ	Stable with BMZ	Progressive Disease	Statistical Analysis
Specific IgE, kU/L	*n* = 37	*n* = 39	*n* = 75	*n* = 7	*X*^2^(3) = 38.564; *p* < 0.001
	M = 1.16 SD = 3.58 MD = 0.12 IQR = 0.42 Min = 0.00 Max = 19.40	M = 1.50 SD = 5.47 MD = 0.14 IQR = 0.81 Min = 0.02 Max = 34.30	M = 4.08 SD = 7.00 MD = 1.57 IQR = 3.85 Min = 0.02 Max = 49.10	M = 5.38 SD = 7.94 MD = 0.69 IQR = 13.53 Min = 0.03 Max = 19.60	
Total IgE, IU/mL	*n* = 37	*n* = 39	*n* = 76	*n* = 7	*X*^2^(3) = 40.856; *p* < 0.001
	M = 100.65 SD = 198.85 MD = 21.50 IQR = 97.85 Min = 3.40 Max = 987.40	M = 516.96 SD = 2313.18 MD = 45.60 IQR = 60.00 Min = 1.60 Max = 14,089.00	M = 580.53 SD = 870.64 MD = 234.65 IQR = 526.57 Min = 1.10 Max = 5251.00	M = 1252.96 SD = 2106.86 MD = 103.00 IQR = 1897.90 Min = 3.80 Max = 5750.00	
IHA IgG, <1:32 = ‘0’	*n* = 37	*n* = 38	*n* = 74	*n* = 7	*X*^2^(3) = 27.233; *p* < 0.001
	MD = 0 P25 = 0 P75 =0 Min = 0 Max = 128	MD = 0 P25 = 0 P75 = 32 Min = 0 Max = 1024	MD = 32 P25 = 0 P75 = 128 Min = 0 Max = 4096	MD = 128 P25 = 0 P75 = 1024 Min = 0 Max = 1024	
Em2^+^ positive	*n* = 33 (89.2%)	*n* = 23 (60.0%)	*n* = 20 (26.3%)	*n* = 1 (14.2%)	Chi-Square-test: *X*^2^(3) = 44.805; *p* < 0.001
Em2^+^ negative	*n* = 4 (10.8%)	*n* = 14 (35.9%)	*n* = 55 (72.4%)	*n* = 6 (85.7%)	
PET negative	/	*n* = 19 (48.7%)	*n* = 8 (10.5%)	*n* = 0 (0%)	Chi-Square-test: *X*^2^(2) = 23.925; *p* < 0.001
PET positive	/	*n* = 20 (51.3%)	*n* = 68 (89.5%)	*n* = 7 (100%)	

**Table 4 pathogens-12-01041-t004:** Multivariate analysis assessing independent risk factors for higher category of clinical outcome. Em2^+^ IgG was excluded due to multicollinearity with IHA IgG. If assessed in a separate model, it did not yield a significant result (*p* = 0.098). Similarly, total IgE was excluded because of multicollinearity with specific IgE. In a separate model, it did not yield significant results (*p* = 0.675). Including Ck18F-M30 and Ck18F-M65 separately instead of the ratio did not yield a significant result (*p*_M30_ = 0.798; *p*_M65_ = 0.756) either.

Variables	Odds Ratio	95% Confidence Interval	*p* =
Gender	0.488	0.198–1.201	0.119
Age	1.031	1.003–1.060	0.029
WHO PNM stage	1.435	0.937–2.195	0.097
PET/CT positivity	8.908	3.019–26.285	<0.001
IHA titers	1.000	0.999–1.001	0.931
Specific IgE levels	1.027	0.950–1.112	0.493
SAA levels	0.980	0.952–1.008	0.165
CRP levels	0.992	0.930–1.059	0.807
sIL-2R levels	1.001	0.998–1.004	0.649
Ratio Ck18F(M30:M65)	0.757	0.427–1.339	0.338
ECP levels	0.978	0.834–1.148	0.787
Absolute EOS count	2.396	0.040–143.45	0.676

## Data Availability

Not applicable.

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
