# Peer review of "18FDG-PET/CT-Scans and Biomarker Levels Predicting Clinical Outcome in Patients with Alveolar Echinococcosis—A Single-Center Cohort Study with 179 Patients"

_pathogens, 2023, doi:10.3390/pathogens12081041_

Round 1

Reviewer 1 Report

Peters and colleagues present new data on biomarker levels and FDG-PET/CT Scans in Alveolar Echinococcosis, which could be used to assess lesional activity. The subject is of high clinical and academic value. The cohort size is quite substantial for an AE study. Overall the manuscript is well written and comprehensible. Regarding the methodology, I see following issues:

- Definition of clinical status: The WHO recommends live-long treatment of inoperable patients with benzimidazoles (BMZ). Here, Peters and colleagues report 42 patients with "stable disease without BMZ". How were these patients selected for a treatment discontinuation? The published protocols combine negative PET-CT with certain neg. serological markers (Em18- or Em2+ antibody). This is very relevant for the interpretation of the presented results, in particular the multivariate analysis.

- Timepoint of PET/CT and biomarker testing. Please state, when and why PET/CT was performed. In cured patients, I presume the timepoint was before surgery. In inoperable patients, was it done to assess patients for treatment discontinuation (i.e. after 2 years of therapy; due to changes in serology) or randomly? Were the biomarkers tested in sera from the timepoint of PET/CT or independently?

- Multivariate analysis: here PET/CT finding (pos. vs. neg.) is the defined outcome/dependent variable. Clinical Status is a defined covariable. As mentioned above, in other publications patients with "controlled disease" are identified through the combination of neg. PET/CT and neg. serological markers. Therefore, this calculation could be intrinsically biased. Depending on whether PET/CT influenced the definition of clinical status, this variable needs to be taken out of the equation. I also highly suggest considering Clinical Status as the dependent variable. This could be shown alone or in addition to the presented calculation - depending on the definition of clinical status.

- Data presentation: given the relative low number of patients per group, data should be presented as median and IQR or range instead of mean and standard deviation. Furthermore, missing data should be included in the percentage calculation (i.e. Table 1: Em2+ positive/negative patients in each group). In Figure 1 I recommend using a different graph format, without connecting lines, since IgE values of independent groups are presented. Finally, I recommend replacing Figure 3 - which only serves an explanatory purpose - with either the first or second data set of the supplements depending on the definition of clinical status (if negative PET is a prerequesite of clinical status, then include the first data set).

Minor remarks:

- Results section:

- line 179: "stage IIa" should probably state "stage IIIa"

Reviewer 2 Report

This study evaluated the value of some biomarkers and 18FDG-PET/CT in the differential diagnosis of patients with AE in terms of cure, stability and progression. The results of this study have some reference value for the clinical monitoring of patients with AE, but there are some defects, which need to be further dealt with.

Minor points:

1. L 76  First, we aimed to identify biomarkers corresponding to the extent and the clinical status of AE.

– This study examined a panel of serological factors, but where are the biomarkers identified? It is an important issue, as the first word of the submission title is <biomarker>, which could not be found in the study conclusion too.

2. L101 …, all AE-patients who received a routine FDG-PET/CT scan over a period of 4 years – what time? 

– Does the “routine” mean periodically or yearly during the 4 years?  

3. L104 at the time of visit-- what time?

4. Figure 1 and Figure 4,

-The Figures did not indicate statistic p value and error bars.

5. Table 1B,in lines Em2+ positive and Em2+negative,

- What was the number of cases of each stage?

Round 2

Reviewer 1 Report

The revisions have improved the manuscript greatly, in particular the changes made to the method section explaining the different situations of treatment discontinuation. The group of patients that were discontinued outside structured discontinuation protocols add special value to this study.